# Estimation of Sodium and Potassium Intake: Current Limitations and Future Perspectives

**DOI:** 10.3390/nu12113275

**Published:** 2020-10-26

**Authors:** Bigina N.R. Ginos, Rik H.G. Olde Engberink

**Affiliations:** 1Department of Nephrology, Amsterdam University Medical Centres, Academic Medical Centre, University of Amsterdam, 1105 AZ Amsterdam, The Netherlands; 2Amsterdam Cardiovascular Sciences, VU University Medical Center, 1081 HV Amsterdam, The Netherlands; r.h.oldeengberink@amsterdamumc.nl

**Keywords:** sodium, potassium, salt, diet, spot urine, 24-h urine, dietary assessment

## Abstract

Globally, average dietary sodium intake is double the recommended amount, whereas potassium is often consumed in suboptimal amounts. High sodium diets are associated with increased cardiovascular and renal disease risk, while potassium may have protective properties. Consequently, patients at risk of cardiovascular and renal disease are urged to follow these recommendations, but dietary adherence is often low due to high sodium and low potassium content in processed foods. Adequate monitoring of intake is essential to guide dietary advice in clinical practice and can be used to investigate the relationship between intake and health outcomes. Daily sodium and potassium intake is often estimated with 24-h sodium and potassium excretion, but long-term balance studies demonstrate that this method lacks accuracy on an individual level. Dietary assessment tools and spot urine collections also exhibit poor performance when estimating individual sodium and potassium intake. Collection of multiple consecutive 24-h urines increases accuracy, but also patient burden. In this narrative review, we discuss current approaches to estimating dietary sodium and potassium intake. Additionally, we explore alternative methods that may improve test accuracy without increasing burden.

## 1. Introduction

Patients with chronic kidney disease (CKD) are strongly advised to reduce their sodium intake, as sodium-induced increases in blood pressure and proteinuria accelerate CKD progression [1,2,3]. However, dietary sodium intake averages around 4 g (g) per day in developed countries, which is twice the maximum daily intake of 2 g as recommended by the World Health Organization (WHO) [4,5]. Reducing sodium consumption has been identified as one of the most cost-effective approaches to lower cardiovascular disease (CVD) incidence and related death [1,6]. Conversely, high potassium intake decreases blood pressure in individuals with hypertension, an effect that may even be strongest at higher levels of sodium consumption [7]. An increasing body of evidence has linked high potassium intake with lower all-cause mortality in CKD populations and a reduced risk for CKD progression and incident CVD [8,9,10]. The WHO recommends a potassium intake of 3.5 g/day. However, in CKD populations, intake is estimated to average around 2.4 g/day [10,11].

Individual health benefits as a result of decreasing sodium intake and increasing potassium intake form a strong incentive for clinicians to guide patients with CKD and/or CVD, without compromised renal handling of potassium, to changing their dietary habits accordingly. Accurate individual-level estimation of sodium and potassium intake is crucial for these personalized dietary recommendations, but is also of importance for investigation of the association between dietary intake and long-term outcome.

A wide range of urinary and non-urinary methods to estimate daily sodium and potassium intake are currently available. The sole use of dietary assessment tools is commonly observed in longitudinal studies, but in interventional research a single 24-h or spot urine collection is predominantly utilized to estimate sodium intake [12]. One of the major concerns in research on dietary sodium intake is the inaccuracy of assessment methods, which has led to conflicting evidence on the optimal sodium intake for the prevention of CKD and CVD [6]. On an individual level, inaccurate estimation of sodium or potassium intake may result in incorrect dietary advice and increase the risk for CKD and CVD.

In this narrative review, we discuss the value of current approaches to estimating dietary sodium and potassium intake on population scale and on an individual level. Additionally, we explore alternative methods that may improve test accuracy without increasing patient burden.

## 2. Dietary Assessment Tools

Non-urinary based dietary assessments methods are broadly used because of their ability to register the intake of a large variety of nutrients at once, reveal potential dietary intervention targets and, depending on the method used, their relatively low costs and participant burden. Although not specifically designed to estimate sodium or potassium intake, dietary assessment tools are being used for this purpose. All dietary assessment tools rely on food databases, thus database inaccuracies (e.g., differences in nutrients between processed, restaurant cooked and home cooked foods) can result in measurement errors. The three most used methods are food frequency questionnaires (FFQs), 24-h diet recalls and diet records (Table 1). FFQs are often used to estimate population intake, diet records to estimate individual intake and 24-h diet recalls have been used for both.

FFQs are checklists composed of predefined foods and beverages, aiming to record consumption frequency within a population during a specified timeframe (weeks or months). The questionnaire takes up 20 to 30 min and can be filled out by participants or by an interviewer. Foods included in each questionnaire vary based on the study population and portion sizes can be defined, although the main focus is often on frequency of consumption [13]. The major upside of FFQs is their ability to screen for a wide variety of foods for relatively low costs and low burden in large groups, but measurement errors can occur if questionnaires are not relevant to local dietary habits or to seasonal changes and due to recall bias [13,14].

Twenty-four hour diet recalls involve documentation of all foods and beverages consumed the previous day. They can be repeated over an extended period of time. They are commonly conducted by a trained interviewer, but an online survey can be used as well. Standard measurements, such as cups and tablespoons, are used to describe quantities. One 24-h diet recall takes 20 to 30 min. This method is prone to recall bias and underreporting [13,15], the latter observed particularly in overweight individuals [16,17]. Twenty-four hour diet recalls may thus be more precise in documenting intake than FFQs, because they are more tailored towards the individual and quantities of consumed foods are defined, but their retrospective nature can cause similar errors.

In contrast to FFQs and 24-h diet recalls, diet records require prospective documentation of intake. Traditionally, records are kept for seven consecutive days, however, keeping records for more than four days was found to give inaccurate results due to respondent fatigue [18]. This method suffers minimally of recall bias, and if foods are documented and weighed correctly it is considered the most accurate dietary assessment tool [13]. Keeping a dietary record is time consuming and has a high burden; individuals need to be literate and trained on how to keep record [13,18]. This method may therefore be less suitable for the CKD population, as limited health literacy and cognitive impairment are common [19,20]. To lower burden, respondents may simplify or underreport complicated meals that they consumed. Further, keeping a diet record may alter eating behaviors in and of itself, such as opting for healthier and more favorable foods [13,15,18]. Dietary records may thus be the most precise in documenting intake and least susceptible to recall bias when compared to FFQs and 24-h diet recalls, but they have a high burden and can misrepresent dietary habits due to behavioral alteration.

### 2.1. Estimation of Average Population Intake

A 2017 systematic review concluded that there was poor agreement between sodium intake estimated by FFQs and 24-h urinary sodium excretion [14]. The investigators were not able to quantitatively compare the studies included in the review due to variability in methods of data collection between validation studies, such as the number of 24-h urine samples collected as a reference method, assessment of the completeness of urine collections and FFQ design. They advised against the use of current FFQs to estimate population sodium intake and called for standardization of validation studies.

A pooled validation study demonstrated that FFQs and repeated 24-h diet recalls are better at estimating potassium intake and dietary sodium to potassium (Na^+^/K^+^) ratio than estimating sodium intake alone [17]. The difference in accuracy between estimates of sodium and potassium intake may be explained by the fact that most sodium in Western diets comes from processed foods, which contain sodium in variable amounts, whereas sources of potassium, like fruits, vegetables, nuts and whole grains contain more consistent amounts of potassium.

### 2.2. Estimation of Individual Intake

Individual-level sodium intake estimates by 24-h diet recalls [15,21] and diet records [15] perform poorly when they are compared to 24-h urine collections as a reference. In a recent study, population averages based on 24-h diet recalls were similar to two 24-h urine collections in individuals from Japan, the United Kingdom and United States, while in China urine estimates were on average 2.3 g higher [21]. However, in 59% of the individual study participants, sodium intake was estimated to be > 2.4 g using 24-h diet recalls, whereas urine-based estimates of the same individuals were < 2.4 g. These data demonstrate that in more than half of the subjects, the dietary advice would depend on the method that is used for estimation of sodium intake. To date, it remains unclear how these methods relate to intake estimates that are based on food ashing.

### 2.3. Dietary Assessment Tools vs. Urine-Based Methods

An important limitation of the research on the value of dietary assessment tools for estimation of sodium or potassium intake is the reference method of 24-h urine collection. Recent data, which will be discussed below, demonstrate major limitations of using 24-h urine collections for estimation of sodium or potassium intake on an individual level when a limited amount of samples are collected. Since only 5 out of 38 of the studies included in two recent meta-analyses used three or more 24-h urine collections as a comparator, interpretation of these data are difficult [14,15]. So far, no studies have compared estimates of sodium and potassium intake by dietary assessment with the actual intake, as measured by food ashing. Adding this reference method will likely improve the quality of studies validating the use of diet records, but will be difficult when validating retrospective methods like FFQs and 24-h diet recalls.

## 3. Twenty-Four Hour Urine Collections

The current standard method for assessing sodium and potassium intake is 24-h urine collection. Individuals are instructed to collect all urine over a period of 24 h, in which sodium and potassium excretion is determined [22]. This method is based on the assumption that 24-h urinary sodium and potassium excretion always reflects 90–95% and 70% of ingested sodium [6] and potassium [11], respectively, given that urine is collected on a representative day and that urine collection is complete. However, recent long-term balance studies demonstrated that the rationale behind this method oversimplifies sodium and potassium balance, and 24-h urine excretion differs significantly from the measured 24-h intake [23,24]. Twenty-four hour urine specimens produce easily quantifiable data, but drawbacks are the high costs, high burden, the recently demonstrated inaccuracies and risk of collection errors.

The gold standard to assess completeness of 24-h urine collection is by administration of para-aminobenzoic acid (PABA) and determination of urine recovery [25]. Other commonly used methods are self-reporting, creatinine index (ratio of observed vs. expected creatinine excretion) and creatinine excretion combined with urine volume, but none of these methods were found to reliably assess the completeness of a 24-h urine collection when compared to PABA recovery [25]. A 2016 systematic review reported that 24-h urine sodium and potassium excretion were on average 0.45 and 0.37 g lower in incomplete collections [26]. These inaccuracies may lead to wrong assumptions and incorrect dietary advice, and may ultimately affect long-term outcome.

### 3.1. Estimation of Average Population Intake

On a population level, under- and overestimations of sodium or potassium intake by 24-h urine collections balance each other out and may thus result in acceptable population estimates. Collections are required to be conducted in representative individuals on representative days. In a Japanese sodium intervention study that compared two rural villages, this approach was able to pick up the effects of a dietary intervention that was only implemented in one of the villages [27]. Further, a recent study found that the use of three 24-h urine collections instead of one did not significantly change estimated mean population sodium intake [28]. These data suggest that 24-h sodium excretion as measured by a single 24-h urine collection is sufficient for estimation of short-term average population intake.

### 3.2. Estimation of Individual Intake

The 24-h urine collection aims to estimate sodium or potassium intake on an individual level to form personalized dietary advice or to investigate the association between dietary intake and long-term health outcomes. Recent long-term balance studies, however, demonstrated that a single 24-h urine collection cannot accurately estimate sodium or potassium intake in healthy volunteers [23,24]. During fixed intake of sodium and potassium in a highly controlled environment, a discrepancy of up to 2 g was observed between 24-h sodium excretion and 24-h intake [24]. Consequently, 51% and 31% of sodium and potassium estimates based on a single 24-h urine collection differed respectively by >0.58 and >0.98 g/day from dietary intake [23].

Discrepancies between intake and excretion were related to infradian cortisol and aldosterone rhythms, which are thought to regulate the storage and release of sodium from a third compartment (Figure 1) [29]. The effect of these infradian rhythms on sodium and potassium balance were minimized by repeating 24-h urine collections [23]. Regardless, when using three consecutive 24-h urine collections, 25% and 18% of sodium and potassium estimates were misclassified (respectively, >0.58 and >0.98 g/day), as well as 8% and 13% when 24-h urine was collected during seven consecutive days. However, collection of seven 24-h urine samples is a high burden for patients.

A major strength of this study is the controlled environment and accurate assessment of sodium and potassium intake. However, this setting limits the generalizability to free-living individuals with a variable intake of sodium and potassium. Furthermore, only young healthy men were included in this study, thus it is unknown whether these infradian rhythms in aldosterone and cortisol affect sodium and potassium balance in women and/or CKD patients to a similar extent.

We have recently demonstrated the consequences of using a single 24-h urine collection for estimation of sodium intake [30]. When estimating dietary sodium intake with multiple 24-h urine collections instead of a single 24-h urine collection, estimated sodium intake was >0.8 g different in half of the subjects (Figure 2B). The fact that estimated population sodium intake was similar with both methods points out that the inaccuracy is particularly found when estimating individual sodium intake (Figure 2A). As a result of the substantial differences in individual sodium intake estimates, hazard ratios for renal and cardiovascular outcome changed by up to 85% when sodium intake estimates were based on multiple 24-urine collections instead of a single one. These data, which were confirmed by others, demonstrate that the methods that are used for sodium intake estimation strongly determine the observed association between sodium intake and long-term CVD and CKD outcomes [31].

Estimation of dietary sodium and potassium intake is often desired for longer periods. This requires multiple measurements at different time points, because sodium and potassium intake are affected by day-to-day variations in diet, medication use and co-morbidities. Nevertheless, most cohort studies only use urine samples from a single point in time for estimation of sodium or potassium intake, whereas the follow-up is often 5–10 years. The use of these inaccurate methods are likely the cause of the conflicting data on the optimal sodium and potassium intake for renal and cardiovascular protection that have been reported [6].

## 4. Spot Urine Based Equations

Investigators have been exploring alternative methods to the 24-h urine collection due to its costs and significant participant burden. As a result, a variety of equations were developed to calculate 24-h sodium excretion, and in some cases also 24-h potassium excretion, from spot urine samples. These equations estimate 24-h creatinine excretion from spot urine creatinine concentrations and different variables such as age, gender, length and height, and correct spot urine sodium or potassium concentration accordingly (Table 2).

Differences between spot urine based equations are primarily the result of differences between the original study populations. The Tanaka [32] and INTERSALT [33] method used data from the INTERSALT study to estimate dietary sodium intake in, respectively, Japanese and Western populations, whereas the CKDSALT method specifically aimed to estimate sodium intake of CKD patients stage 1–4 [34]. The Mage method was originally not designed to estimate sodium intake, but for estimation of pesticide exposure in the National Health and Nutrition Examination Survey (NHANES) cohort [35,36].

Spot urine equations other than the Kawasaki (second morning) [37] and CKDSALT (morning) [34] method allow for random spot urine sampling. Collection of spot urine has a negligible risk of collection errors and is less burdensome and more affordable than 24-h urine collection, which makes this method appealing for use in research. However, it should be emphasized that these equations were developed to estimate 24-h sodium and/or potassium urinary excretion, which was recently shown to be an inaccurate estimate of 24-h intake [23,24]. Thus, one may argue that all spot urine based equations are fundamentally flawed, as they are based on this false assumption.

### 4.1. Estimation of Average Population Intake

Many spot urine equation validation studies have been conducted with varying results in different populations. For instance, among healthy young adults in the United States, the INTERSALT method provided the best estimation of 24-h sodium excretion [35], and in Chinese and multi-ethnic populations the Kawasaki method was superior [39,40,41]. Nevertheless, all commonly used equations display systematic overestimation at low 24-h sodium excretion levels and underestimation at high excretion levels [42,43]. When validating equations for estimation of 24-h potassium excretion, some studies demonstrated that the Kawasaki method provided the least biased estimation [40,41], while others found no difference between the Kawasaki, Tanaka and Mage method [44].

The major limitation of these validation studies, however, is that most studies lack a proper measure of dietary intake, such as repeated 24-h urine collection or measured intake through a standardized diet, but used dietary assessment tools or a single 24-h urine collection as a comparator [6]. Inconsistencies in current evidence have led the International Consortium for Quality Research on Dietary Sodium/Salt to conclude that more high-quality validation studies are needed to assess whether spot urine based equations are suited for estimating population average sodium intake [6].

### 4.2. Estimation of Individual Intake

Spot urine based equations are in principle designed to estimate the average population intake, but are often used to estimate individual intake when resources are limited. Several validation studies have demonstrated poor performance in estimating both sodium and potassium intake on an individual level [42,44,45]. One validation study that specifically examined the use of spot urine based equations to estimate individual sodium intake found that 63% (Kawasaki), 79% (INTERSALT) and 66% (Tanaka) of the participants were misclassified (sodium intake < 3.6 g/day vs. 3.6–4.8 g/day vs. 4.8–6 g/day vs. >6 g/day) when spot urine sodium estimates were compared to 24-h sodium excretion [45]. Moreover, most validation studies have been conducted in healthy participants and do not take into account the potential effects of disease and medication use on sodium and potassium excretion. CKDSALT investigators did attempt to incorporate these effects by studying a population with CKD, but their method has a major limitation: the equation requires measurement of 24-h urine volume (Table 2) [34]. This, consequently, requires the patient to collect urine for 24 h, which defeats the purpose of using a spot urine based equation for estimating sodium intake.

To our knowledge, all published studies investigating spot urine based equations compare sodium and/or potassium estimates from spot urine samples to 24-h excretion. However, due to the significant shortcomings of 24-h urine collection as a method to estimate sodium and potassium intake, it should not be used as the main reference method. Future studies should therefore compare the performance of spot urine equations to the measured dietary intake.

## 5. CKD Patients

Although CKD patients are an important target group for dietary sodium and potassium assessment, few studies investigating methods for sodium and potassium intake estimation have focused on these patients. As dietary assessment tools are not affected by changes in sodium and potassium homeostasis, these may be useful for estimation of sodium and potassium intake as well as assessment of overall nutritional status [46]. However, besides the previously discussed limitations, the use of dietary assessment tools can be complicated by the limited health literacy and cognitive impairment that is present in CKD populations, especially when using weighed diet records [19,20].

Urine-based methods face different challenges. In CKD patients, sodium and potassium homeostasis is often altered as a result of reduced glomerular filtration rate, co-morbidities such as hypertension, and the use of medication such as diuretics, angiotensin-converting enzyme inhibitors and angiotensin receptor blockers [47]. On top of that, the use of spot urine samples is complicated by changes in circadian sodium excretion rhythms that occur with decreasing kidney function [48]. When the capacity for daytime renal sodium excretion is reduced, blood pressure and sodium excretion will increase at night. These changes have not been taken into account in the formulas that are used to estimate 24-h sodium excretion from spot urine samples. This limitation is illustrated by a recent trial that demonstrated that four spot urine based equations misclassified individual sodium intake, as measured by an average of 3.5 24-h urine collections, in more than half of the subjects [49]. As discussed previously, the usefulness of the recent CKDSALT spot urine equation is limited by the need for measurement of 24-h urine volume. Other spot urine based equations that were developed to use in CKD populations have neither been sufficiently externally validated nor did they compare sodium intake estimates to measured dietary intake [50,51,52,53].

Collection of 24-h urine samples is not affected by circadian rhythms in sodium excretion. However, in CKD patients, the use of 24-h urine collections for sodium and potassium intake estimation has not been compared to the measured dietary intake. It is therefore unknown whether 24-h urine collection is a valid method in the CKD population. The recently described aldosterone- and cortisol-induced rhythmic variations in sodium and potassium excretion have not been investigated in CKD patients. Studies that validate the use of 24-h urine collections for estimation of sodium and potassium intake in CKD populations are needed.

## 6. Practical Implications

Dietary assessment tools are insufficiently validated for estimation of average population and individual-level dietary sodium and potassium intake.Spot urine based equations have substantial limitations when estimating average population sodium and potassium intake, and should not be used for estimation of individual-level intake.A single 24-h urine collection can be used to estimate short-term average population sodium or potassium intake.Multiple 24-h urine collections are needed to estimate individual-level sodium and potassium intake. For estimation of individual-level intake on the long term, this set of measurements should be repeated.The use of spot urine samples and 24-h urine collections for estimation of sodium and potassium intake has been insufficiently validated in CKD patients.

## 7. Potential Future Methods

### 7.1. Repeated Spot Urine Based Equation

The observation that the accuracy of individual sodium and potassium intake estimates increases when seven consecutive 24-h urine collections are averaged suggests that the quantity of measurements may be more important than their quality [23]. In this respect, repeated collection of spot urine samples may be an alternative, with fair accuracy and low patient burden. A study that investigated the potential value of repeated spot urine collection for estimation of average population and individual-level sodium intake found that accuracy of estimates increased with the addition of extra samples [54]. Concordance correlation coefficients were 0.20 (one spot urine sample), 0.31 (two spot urine samples) and 0.42 (three spot urine samples). Another study examined the value of repeated spot urine samples to estimate average population sodium intake, but found no improvement when three samples were used [28]. Both studies aimed to estimate 24-h sodium excretion as measured by ≤3 24-h urine collections. As discussed above, this reference method has significant limitations. So far, it is unknown how sodium and potassium estimates from repeated measurements of spot urine samples relate to the actual dietary intake.

### 7.2. Urinary Sodium to Potassium Ratio

Daily sodium and potassium excretion are often measured separately, but, alternatively, the urinary sodium to potassium (Na^+^/K^+^) ratio could be calculated, as renal handling of potassium and sodium are highly interconnected. When potassium intake is high, it increases natriuresis by modulating the sodium-chloride cotransporter in the distal convoluted tubule [55]. High sodium intake may thus differently affect blood pressure and long-term outcome when combined with high or low potassium intake [7]. According to the WHO, the optimal dietary Na^+^/K^+^ ratio is <1 mmol/mmol (0.59 g/g), and after correction for extrarenal potassium loss, the ideal urinary Na^+^/K^+^ ratio would be <1.3 mmol/mmol (0.77 g/g) [11].

The urinary Na^+^/K^+^ ratio can be quantified in a 24-h urine collection or spot urine sample. An advantage of measuring the 24-h Na^+^/K^+^ ratio is the fact that the ratio is less affected by collection errors than separate measurements of 24-h sodium and potassium excretion. A benefit of the spot urine Na^+^/K^+^ ratio is that no conversion into 24-h excretion values is required, a calculation that adds imprecision. These benefits are likely to increase the accuracy and will facilitate its use in daily clinical practice. Nonetheless, similar to measuring sodium and potassium separately in spot urine samples, the accuracy of the Na^+^/K^+^ ratio is limited by diurnal variation in sodium and potassium excretion (Figure 3).

Another benefit of the Na^+^/K^+^ ratio is that it incorporates the protective effects of both low sodium and high potassium intake in one parameter. Cohort studies have demonstrated that the dietary Na^+^/K^+^ ratio, as estimated by 24-h dietary recalls or FFQs, is more strongly related to CVD mortality and CKD risk than sodium or potassium intake alone [57,58]. For example, in the Tehran Lipid and Glucose Study, a high estimated dietary Na^+^/K^+^ ratio (2.43 vs. 0.61) was associated with an increased risk of CKD (odds ratio 1.52, 95% CI 1.01–2.30), whereas estimated sodium or potassium intake alone were not associated with CKD risk [57].

Cohort studies that used urine collections for estimation of dietary Na^+^/K^+^ ratio, however, show contrasting results. Data from the Prevention of Renal and Vascular End-stage Disease (PREVEND) study, in which dietary intake was estimated with four 24-h urine collections, showed no association between urine Na^+^/K^+^ ratio and long-term renal or cardiovascular outcomes, whereas low potassium but not high sodium excretion was associated with increased risk of CKD [8,9]. Yet, an increased urine Na^+^/K^+^ ratio, as measured by a single spot urine sample, was associated with increased risk of stroke in a multi-ethnic population from the United States [59]. Thus, more high-quality research is needed to establish the true association between the dietary Na^+^/K^+^ ratio and long-term CVD and CKD.

Data on how the dietary Na^+^/K^+^ ratio can be best estimated are scarce, as no studies have compared estimates from dietary assessment tools or urine collection with the actual dietary intake. A study from Japan suggests that repeated collection of spot urine samples may be used for estimation of dietary Na^+^/K^+^ ratio [60]. In healthy volunteers, the mean urine Na^+^/K^+^ ratio of six randomly collected spot urine samples was strongly correlated with the reference method, seven consecutive 24-h urine collections. The performance of six randomly collected spot urine samples was comparable to two 24-h urine collections. Similar data were found in hypertensive subjects with and without antihypertensive treatment [61]. Studies that track both sodium and potassium intake and urinary excretion in healthy individuals and individuals with CKD are needed to reveal more details on how the urinary Na^+^/K^+^ ratio can best be determined.

## 8. Conclusions

Current available methods to estimate individual sodium and potassium intake either lack accuracy or have a high patient burden. For assessment of population sodium and potassium intake, a single 24-h urine collection can be used, but multiple 24-h urine collections are needed when estimating individual intake or when these estimates are related to individual outcome data. High-quality studies that focus on increasing accuracy or lowering costs and burden are needed. The performance of repeated spot urine sampling or measurement of urinary Na^+^/K^+^ ratio for estimation of sodium and potassium intake needs to be investigated. When investigating new estimation methods, sodium and potassium intake estimates should be compared to measured dietary intake to ensure the use of a valid reference method.

## Figures and Tables

**Figure 1 nutrients-12-03275-f001:**
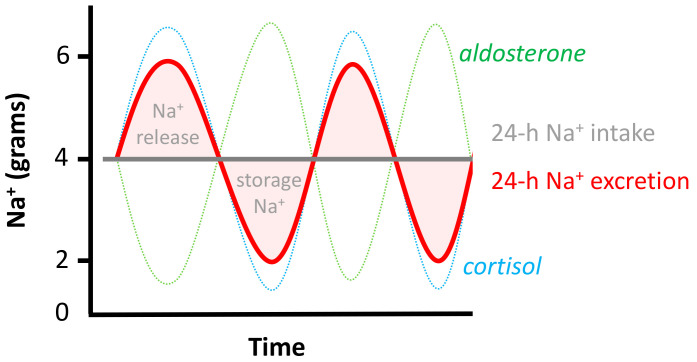
Sodium storage and release from a third compartment. Twenty-four hour sodium (Na^+^) excretion is substantially different from the 24-h intake during fixed Na^+^ intake due to the release and storage of Na^+^ from/in a third compartment, which correlates with infradian rhythms of cortisol and aldosterone.

**Figure 2 nutrients-12-03275-f002:**
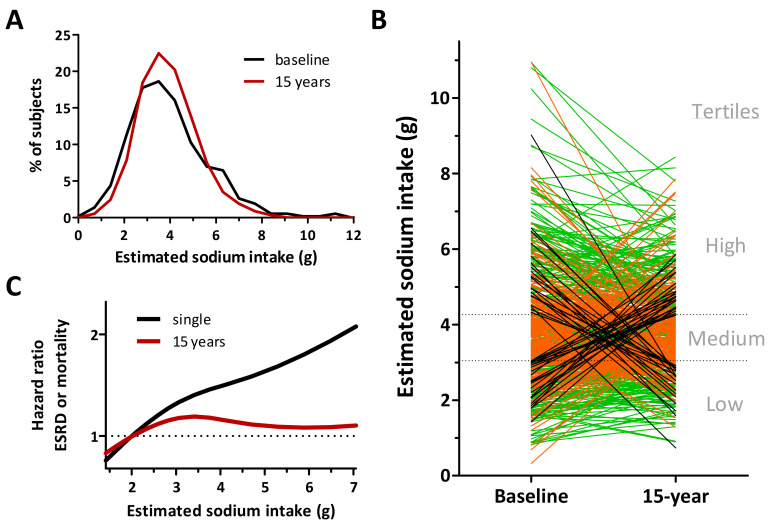
Single versus multiple 24-h collections. We estimated sodium intake with a single baseline collection and averaged collections during 15-year follow-up. (**A**) Population sodium intake was similar when estimated at baseline or during the 15-year follow-up. (**B**) However, individual level estimates of sodium intake were substantially different when multiple measurements were taken into account. Only 50% of the subjects remained in the similar sodium intake tertile (green lines), while 42% moved to the adjacent tertile (orange lines) and 8% of the subjects switched between the outer tertiles (black lines). (**C**) When investigating the association between long-term renal outcome and sodium intake, hazard ratios changed significantly when baseline sodium intake estimates were used instead of 15-year estimates. Adjusted from [30].

**Figure 3 nutrients-12-03275-f003:**
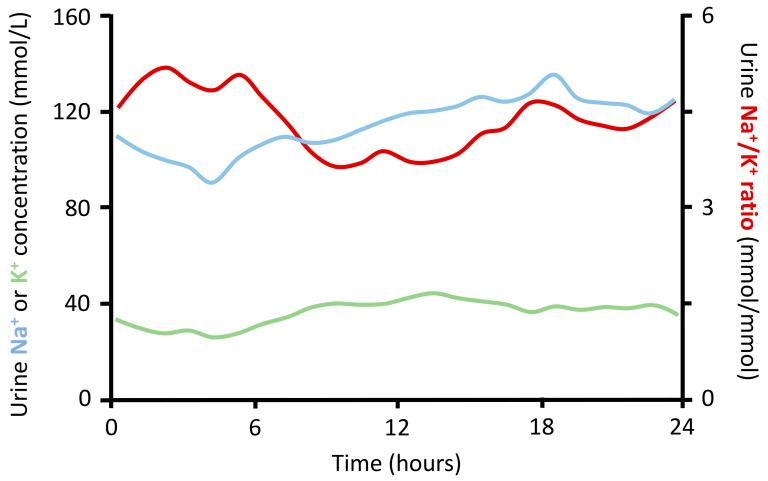
Diurnal variation in sodium and potassium excretion. Sodium (Na^+^) and potassium (K^+^) excretion varies from hour to hour. As a result, the urine sodium to potassium (Na^+^/K^+^) ratio depends on the moment of urine sample collection. These data are based on 13,277 urine samples from 122 participants with and without hypertension [56]. No difference in Na^+^ or K^+^ excretion or urine Na^+^/K^+^ ratio was found between normotensive and hypertensive individuals. Adjusted from Iwahori et al. [56].

**Table 1 nutrients-12-03275-t001:** Strengths and limitations of commonly used dietary assessment methods for estimation of sodium and potassium intake.

	FFQ	24-h Diet Recall	Diet Record
Costs	low	moderate	moderate to high
Time and burden	low	low to moderate	high
Precision	low	moderate	high
Timeframe	weeks or months	one day, but can be repeated over weeks or months	days
Biases	recall bias, information bias	recall bias	social desirability, selective reporting

Food frequency questionnaires (FFQ) are the most cost-effective and timesaving, but have low precision. Main biases stem from respondents not correctly memorizing their intake or questionnaires not being relevant to local dietary habits. Diet records have the highest precision, but are costly, burdensome and time consuming. Reporting may be biased due to change of dietary habits as a result of keeping a diet record or omitting a report of complicated foods due to high burden. Twenty-four hour diet recalls sit somewhere in between FFQs and diet records in terms of precision, costs, time and burden. Results are mainly affected by recall bias, but social desirability and selective reporting could also play a role.

**Table 2 nutrients-12-03275-t002:** Overview of commonly used spot urine based equations.

	Age	Sex	Height	Weight	BMI	Spot Na^+^	Spot K^+^	Spot Cr	Spot Ur	24-h uV						
Method	Equation Variables										Region	Men	Women	Age	BMI	eGFR
INTERSALT [33]							■				North America and Europe	2841	2852	20–59	25.8	N/A
CKDSALT [34]											China	2939	2296	54	24.2	56.7
Kawasaki [37]											Japan	78	81	34	22.1 *	N/A
Tanaka [32]											Japan	295	296	40	22.4	N/A
Toft [38]											Denmark	102	371	51	25.5	N/A
Mage [36]											United States	483	246	N/A	N/A	N/A

Grey marked variables are included in the method equation: age, sex, height, weight, body mass index (BMI), spot urine sodium concentration (spot Na^+^), spot urine potassium concentration (spot K^+^), spot urine creatinine concentration (spot Cr), spot urine urea concentration (spot Ur) and 24-h urine volume (24-h uV); ■ indicates that the use of spot K^+^ for estimation of sodium intake is optional. Summary of original study population characteristics per method: age (years), BMI (kg/m^2^) and eGFR (ml/min/1,73 m^2^) are displayed as mean, median or range; * calculated mean BMI using provided mean height and weight.

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
