# Peer review of "Estimation of Sodium and Potassium Intake: Current Limitations and Future Perspectives"

_nutrients, 2020, doi:10.3390/nu12113275_

Round 1
Reviewer 1 Report
This is a useful narrative review regarding the estimation of sodium and potassium intake from urine based measures.
I have some comments (in the order they appear in the manuscript):
Line 46: ('..improve overall health'). I think it is more precise to say cardiovascular disease.
Line 57: As this is not a systematic review, I suggest adding narrative (In this narrative review…)
The interpretation of Table 1 is somewhat difficult. Usually, +++ is better than +, but for costs, it seems to be the other way around. In addition, subject bias should be defined.
Line 78/79: In many instances FFQ also collect information on portion size, not only frequency
Line 81: Better information than merely ranking intake high and low intake can be achieved in many instances.
Line 112-114: Better compared to what?
Line 125:’These data demonstrate that in more than half of the subjects the dietary advice would depend on the method that is used for estimation of sodium intake. To date, it is unclear which method is the right one.’ Comment: I suggest reformulating this sentence in order to make the meaning more clear.
Line 132: I suggest including ‘on an individual level’ to ‘demonstrate major limitations of using 24-hour urine collections for 131 estimation of sodium or potassium intake..’
Line 133: Seven 24-hour urine collection is a lot. How well do we know that? In Cobb, L.K. et al. Methodological issues in cohort studies that relate sodium intake to cardiovascular disease outcomes: a science advisory from the American Heart Association. Circulation 2014, 129, 1173–1186, more than four collections are mentioned.
Line 143: To be more precise, the assumption is that it reflects approximately 90% of dietary intake is excreted in the urine.
Line 167-177: This information is interesting. According to ref. 21, the information comes from a simulated trip to Mars. How generalizable are these finding for the general population? In the article, it is stated: ‘The number of subjects was relatively small. We studied only men, and the responses in women could be different. It is possible that volunteers for a Mars flight simulation are not representative of the general population, although we do not believe that state of affairs to be a confounder in this study’.
Line 198-208 refer to figure 1. However, there are currently two figure 1 in the manuscript. In addition, the one presented on page 6 is somewhat confusing. When did the 24-hours urine collection take place?. If it took place at baseline and after 15-years (panel B in the figure), it is not surprising that the participants changed category.
Line 213: I think it is correct to add ‘spot’ to ‘Most cohort studies have used a single spot urine collection’
Line 299: According to what is stated earlier in the manuscript concerning spot urine, ‘insufficiently validated’ could be made clearer’ (conf. line 273: poor performance in estimating both sodium and potassium intake on an individual level)
Line 366: It could be added that a single 24-hour urine collection can be used to estimate short-term average population intake.
Author Response
Dear Editors,
Please find enclosed the revised manuscript for further consideration for publication in Nutrients. We would like to thank the reviewers and editorial team for their valuable comments and suggestions. We have adapted the article accordingly and feel that the comments have improved the article. Below we will proceed with a point-by-point discussion of the comments of the reviewers. Our reactions are presented in blue. Citations from the article are provided in italic, showing changes to the original article marked in red. Deleted passages are crossed out.
We have attached both the manuscript with highlighted and crossed out changes as well as the final manuscript.
All the authors have read and approved the submitted version of the manuscript you have received. We look forward to your further consideration.
Yours sincerely, on behalf of all authors,
Bigina Ginos, MD
Rik Olde Engberink, MD PhD
Reviewer: 1
This is a useful narrative review regarding the estimation of sodium and potassium intake from urine based measures. I have some comments (in the order they appear in the manuscript):
Line 46: (‘...improve overall health'). I think it is more precise to say cardiovascular disease.
Line 57: As this is not a systematic review, I suggest adding narrative (In this narrative review…)
We want to thank the reviewer for these remarks. We adjusted the sentences accordingly.
Line 46
‘On a population scale, reducing sodium consumption has been identified as one of the most cost-effective approaches to improve reduce overall health CVD incidence and CVD related death.’
Line 58
‘In this narrative review, we discuss the value of current approaches to estimating dietary sodium and potassium intake on population scale and on an individual level.’
The interpretation of Table 1 is somewhat difficult. Usually, +++ is better than +, but for costs, it seems to be the other way around. In addition, subject bias should be defined.
We agree with the comments of the reviewer on table 1. We replaced the plus signs with the terms ‘low’, ‘moderate’ and ‘high’ in the table to avoid confusion (line 72). We replaced ‘subject bias’ by ‘social desirability’ and ‘selective reporting’ and added an extra row to the table to describe the timeframe in which each dietary assessment can take place. These adjustments allowed us to make the current figure description more concise and elaborate on the biases.
Line 74
‘Food frequency questionnaires (FFQ) are the most cost-effective and time saving, but saving but have low precision, as quantities are not registered. Main biases stem from respondents not correctly memorizing their intake or questionnaires not being relevant to local dietary habits. Diet records have the highest precision, but are the most costly, burdensome and time consuming, as participants need to weigh and register all consumption and receive training on how to do this beforehand, but they allow for detailed assessment of intake. Reporting may be biased due to change of dietary habits as a result of keeping a diet record or omitting to report complicated foods due to high burden. 24-hour diet recalls take up more time than answering FFQs, as they are usually conducted over multiple days, but less than keeping diet records. Quantities of consumed foods are estimated in standard measurements (teaspoons, tablespoons, cups etc.), which improves their precision. Costs are on the higher side, as intake is usually assessed by a trained interviewer sit somewhere in between FFQs and diet records in terms of precision, costs, time and burden. Results are mainly affected by recall bias, but social desirability and selective reporting could also play a role.’
Line 78/79: In many instances FFQ also collect information on portion size, not only frequency
Line 81: Better information than merely ranking intake high and low intake can be achieved in many instances.
We would like to thank to thank the reviewer for pointing out that FFQs also collect information on portion size. We removed this information from the description from table 1 as is stated above.
We mentioned that intake is ranked as high or low, because for many (micro)nutrients numerical intake cannot be estimated with a high degree of certainty, when using an FFQ, due to the variability of nutrients in processed foods. However, we realize that this limitation applies to all dietary assessment tools, and is not specific to the FFQ design, thus we removed that intake is merely ranked by high and low.
Line 90
Results are analyzed with a food composition database, ranking intake as high or low. Foods included in each questionnaire vary based on the study population and portion sizes can be defined, although the main focus is often on frequency of consumption.
Line 112-114: Better compared to what?
The cited study demonstrated that estimations of potassium intake and the Na+/K+ ratio were more accurate than estimates of sodium intake alone. We adjusted the sentence as stated below in order to clarify this.
Line 127
‘A pooled validation study demonstrated that FFQs and repeated 24-hour diet recalls are better at estimating potassium intake and dietary sodium-to-potassium (Na+/K+) ratio than estimating sodium intake alone.’
Line 125: ‘These data demonstrate that in more than half of the subjects the dietary advice would depend on the method that is used for estimation of sodium intake. To date, it is unclear which method is the right one.’ Comment: I suggest reformulating this sentence in order to make the meaning more clear.
Line 132: I suggest including ‘on an individual level’ to ‘demonstrate major limitations of using 24-hour urine collections for 131 estimation of sodium or potassium intake.’
Line 133: Seven 24-hour urine collection is a lot. How well do we know that? In Cobb, L.K. et al. Methodological issues in cohort studies that relate sodium intake to cardiovascular disease outcomes: a science advisory from the American Heart Association. Circulation 2014, 129, 1173–1186, more than four collections are mentioned.
We want to thank the reviewer for these observant remarks, we adjusted the sentences accordingly. In a recent position paper of the International Consortium for Quality Research on Dietary Sodium/Salt on the use of urine collections for assessment of dietary sodium intake (J Clin Hypertens. 2019, 21, 700-709), it is stated that ‘The required number of 24-hour urine collections to obtain a stable estimate of usual sodium intake is likely to differ with different dietary patterns, populations, and settings, and has estimated to be at least 3 non-consecutive days’. This position paper refers to a paper that demonstrates that 7 consecutive 24-hour urine collections result in more precise estimated compared to 3 consecutive 24-hour urine collections. The exact amount of necessary collections is unknown but it is likely that the 7th urine collection will have less added value than the 3rd urine collection. We have therefore removed the number ‘7’ from the sentence and discussed the value of 3 vs 7 consecutive 24h urine collection later in the manuscript.
Line 140
‘To date, it is remains unclear which how these methods is the right one relate to estimates based on food ashing.’
Line 144
‘Recent data, which will be discussed below, demonstrate major limitations of using 24-hour urine collections for estimation of sodium or potassium intake on an individual level when less than 7 a limited amount samples are collected. As none only 5 out of 38 of the studies included in two recent meta-analyses used 73 or more 24-hour urine collections as a comparator, interpretation of these data are difficult.’
Line 209
‘Regardless, when using 3 consecutive day urine collections, 25% and 18% of sodium and potassium estimates were misclassified (respectively >0.58 and >0.98 g/day different from the fixed dietary intake), whereas 8% of sodium and 13% of sodium and potassium estimates based on 7 consecutive day collections were misclassified. respectively >0.58 and >0.98 g/day different from the fixed dietary intake. Further, obtaining However, increasing the number of 7 24-hour urine collections from 3 to 7 involves high costs and patient burden, and is not likely infeasible in both clinical and investigative context.’
Line 143: To be more precise, the assumption is that it reflects approximately 90% of dietary intake is excreted in the urine.
Line 156
‘The assumption behind This method is based on the assumption that 24-hour urinary sodium and potassium excretion always reflects 24-hour dietary intake 90-95% and 70% of ingested sodium and potassium, given that urine is collected on a representative day and that urine collection is complete. However, recent long-term balance studies demonstrated that the rationale behind this assumption is false method oversimplifies sodium and potassium balance and 24-hour urine sodium and potassium excretion differs significantly from the measured 24-hour intake.’
Line 167-177: This information is interesting. According to ref. 21, the information comes from a simulated trip to Mars. How generalizable are these finding for the general population? In the article, it is stated: ‘The number of subjects was relatively small. We studied only men, and the responses in women could be different. It is possible that volunteers for a Mars flight simulation are not representative of the general population, although we do not believe that state of affairs to be a confounder in this study’.
We agree that with the reviewer that the results from the simulated Mars trip study may not be generalizable to the population at large. We did already bring up that the results may not be generalizable to the CKD population, but we will expand this further to the general population and point out that this is a limitation.
Line 216
‘A major strength of this study is the controlled environment and diet: it provides a valid high-quality reference standard for sodium and potassium intake. However, this setting limits the generalizability to free living individuals with a variable intake of sodium and potassium. Furthermore, only young healthy men were included in this study, thus More importantly, it is currently it is unknown whether these infradian rhythms in aldosterone and cortisol affect sodium and potassium balance in women and/or CKD patients to a similar extent.
Line 198-208 refer to figure 1. However, there are currently two figure 1 in the manuscript. In addition, the one presented on page 6 is somewhat confusing. When did the 24-hours urine collection take place? If it took place at baseline and after 15-years (panel B in the figure), it is not surprising that the participants changed category.
We want to thank the reviewer for pointing this out. Two figures were named figure 1, we changed one into figure 2 accordingly.
In the study that is discussed sodium intake was estimated with a single baseline collection and the average of samples collected during a 1-, 5-, and 15-year follow-up. We added this information to the description of the figure.
In figure 2A is displayed how estimated population sodium intake is similar when comparing 24-hour urine collected at baseline to the average of samples collected during 15-year follow up. Figure 2B then shows that this does not automatically translate to the same conclusions on an individual level, as only 50% of the subjects remained in the similar sodium intake tertile. Sodium intake estimates changed with the addition of extra 24-hour urine collection, even within a year of follow-up. These changes can be explained both by moderate reproducibility (change of estimated sodium intake with a follow-up <1 year) and change in dietary pattern on the long term (change of estimated sodium intake with a follow-up 15 year). The reviewer is right to point out that it is not surprising that the participants changed sodium intake category, and this is an important problem with previous studies. Many studies have estimated sodium intake once at baseline whereas the follow-up is 5-10 years without considering changes in sodium intake during this follow-up.
Line 225
‘When estimating dietary sodium intake with multiple 24-hour urine collections instead of a single 24-hour urine collection, estimated sodium intake was >0.8 g different in half of the subjects (Figure 21B). The fact that estimated population sodium intake was similar with both methods, points out that the inaccuracy is particularly found when estimating individual sodium intake (Figure 21A).’
Line 243
‘We estimated sodium intake with a single baseline collection and averaged collections during 1-, 5- and 15-year follow up. (A) Population sodium intake was similar when estimated at baseline or during 15-year follow-up. (B) However, individual level estimates of sodium intake were sodium intake estimates changed substantially different when multiple measurements were taken into account. Only 50% of the subjects remained in the similar sodium intake tertile (green lines) while 42% moved to the adjacent tertile (orange lines) and 8% of the subjects switched between the outer tertiles (black lines). (C) When investigating the association between long-term renal outcome and sodium intake, hazard ratios changed significantly when baseline sodium intake estimates were used instead of 15-year estimates.’
Line 213: I think it is correct to add ‘spot’ to ‘Most cohort studies have used a single spot urine collection’
We did not differentiate between 24-hour and spot urine collection, because we wanted to emphasize the use urine samples at only one timepoint, whereas the data displayed in figure 2 demonstrates that multiple measurements at different timepoints are required. We changed the statement to clarify this.
Line 236
‘Nevertheless, most cohort studies have used a single urine collection only use urine samples from a single point in time for estimation of sodium or potassium intake whereas follow-up is often 5-10 years.’
Line 299: According to what is stated earlier in the manuscript concerning spot urine, ‘insufficiently validated’ could be made clearer’ (conf. line 273: poor performance in estimating both sodium and potassium intake on an individual level)
We want to thank the reviewer for pointing this out. We rephrased this sentence to make it more clear. This section of the paper is intended to give a concise overview of the performance of each estimation method. We elaborate on the reasoning behind these statements in the paragraphs that are dedicated to discussing each method individually.
Line 325
Spot urine based equations have substantial limitations when estimating average population sodium and potassium intake and are insufficiently validated should not be used for estimation of individual-level intake.
Line 366: It could be added that a single 24-hour urine collection can be used to estimate short-term average population intake.
We want to thank the reviewer for this remark, we added this to our conclusion.
Line 393
“Current available methods to estimate individual sodium and potassium intake either lack accuracy or have a high patient burden. For assessment of population sodium and potassium intake a single 24-hour urine can be used, but multiple 24-hour urine collections are needed when these estimates are related to individual outcome data. High quality studies that focus on increasing accuracy or lowering costs and burden are needed. When investigating new estimation methods, sodium and potassium intake estimates should be compared to measured dietary intake to ensure the use of an adequate a valid reference method.”
Reviewer: 2
Some reviews already discussed about the difference in risk of Na consumption on CVD mortality or incidence in which Na consumption was measured by 24hour urines or spot urines shown in epidemiological cohort studies. In my understanding, there have been few manuscript discussing preferred methods focusing urinary measurement, rather than dietary survey methods, to estimate Na and K intakes in clinical settings giving dietary advices.
In this current manuscript, authors seems to try to focus on better assessment method to use in those clinical settings although it is not clear in the title. But it is obvious from the abstract in which authors emphasize the importance of the accuracy of Na intake estimation and without it, clinicians and dieticians may provide misdirected advices. And they compared Na and K estimates obtained from various urine specimens and various dietary surveys.
If that (what is the preferable method to estimate Na and K in clinical settings) is the idea of the authors, my idea is that the manuscript lacks requisite considerations.
WHO’s recommendation of salt intake (less than 5g/day) or recommendations by hypertension societies (less than 6g/day in many case) are well below the average habitual consumption level in most of the civilized society with the exception of minority people who do not have habit of use of salt. It is obvious if one see the foods and menus consumed in the countries. So, dieticians working in clinical settings make dietary advises in the assumption that the current habitual salt intake level would not be lower than the salt reduction target values. I think it is not harmful for the patients, because the target values are not upper limit of requirement but upper limit of harmful level.
In clinical settings, it is more important that dieticians can make advises knowing what are the main source foods of Na or K in the patient’s diet and their contributions to the daily intake. Without it, dieticians cannot make suitable advice. In this point, dietary surveys (FFQ, 24hour dietary recall, or dietary records) have significant advantages over estimates using urine specimen. Authors do not discuss about this point, therefore, I’m afraid clinicians and dieticians who seek for better Na or K estimate method for use in dietary education settings may not find useful knowledge from this manuscript. They would want methods in which they can guess the salt intake trend of a patient, upward or down ward, and the contributing foods.
As a whole, authors discussed about “accuracy” of Na intake estimation derived from various urine specimen. Discussions over effect of salt reduction by diet education programs using various urine or dietary assessment, if they wanted to discuss on the preferred estimation method.
We would like to thank the reviewer for the observant commentary. In our manuscript we describe all available methods to estimate sodium and potassium intake. We discuss characteristics, strengths and limitations of each method so the reader can assess whether the method is suitable for their patient and/or study population. We also give examples of how limitations translate to practice, e.g.: keeping a diet record may not be suitable for populations with cognitive impairment (line 111) or the high cost and burden of repeating 24-hour urine collection 7 times is not feasible in clinic and in most large study populations (line 213). However, our main aim is to give the reader the big picture behind sodium and potassium intake assessment methods: explain how each method builds on the other one and why new insights that question the accuracy of a 24-hour urine collection also affects interpretation of the other methods.
We agree with the reviewer that in a clinical setting it is not harmful to set sodium (and potassium) intake targets below what is recommended by the WHO. However, when the effects of dietary interventions are assessed it can be discouraging for a patient to receive a falsely inflated sodium intake estimate due to measurement errors or infradian variation, as they are often putting in a lot of effort to change their diet (for many this will mean eating more home cooked meals with less processed ingredients). We think it will be easier for patients to adhere to successful dietary interventions if estimated intake reflects real intake and, conversely, it will be easier for dieticians to detect non-adherence. We did not discuss the absolute level of salt or potassium intake recommendation because it is beyond the scope of this review.
The reviewer justly points out that a strength of dietary assessment tools is its ability to provide both intake estimates and targets for dietary intervention. We will include this in the review. This practical benefit does, however, not outweigh the importance of correctly assessing whether a patient eats too much sodium and needs to change their diet. Especially considering the time and effort it takes to successfully carry out dietary interventions. As pointed out in the paper, the accurateness of dietary assessment tools is limited, and they are susceptible to bias. Weighted diet records are the most capable of giving an acceptable estimate of sodium (and potassium) intake, but are costly, burdensome and highly susceptible to measurement errors if respondents have cognitive impairments or are not properly trained. Instead, one may suggest that after assessing that dietary intervention is needed, using a more accurate and lower burden urine based method, dieticians instruct patients to keep a food journal (e.g. using photos) to identify potential targets.
Line 62
‘Non-urinary based dietary assessments methods are broadly used because of their ability to register the intake of a large variety of nutrients at once, reveal potential dietary intervention targets and, depending on the method used, their relatively low costs and participant burden.’
Minor points
L46 “Accurate estimation”
I’m afraid authors use the word “accurate” in a rather crude manner. I think the accuracy desired in the clinical settings would be how the habitual salt intake level of a participant can be estimated.
We agree with the reviewer that in a clinical setting sodium (and potassium) intake estimation does not need to be narrowed down to minor discrepancies (e.g. less than 100 mg difference between estimated and real intake). However, the methods that we discuss in the manuscript produce estimates that can differ up to multiple grams from real intake. These are clinically relevant inaccuracies, especially when the effects of dietary interventions are assessed. We discuss per method to what extend accuracy is compromised so the reader can judge whether these discrepancies between estimated and real sodium (and potassium) intake are acceptable for their patient and/or study population.
L60 “Dietary assessment tools”
Food data base is the important point in assessing Na intake from dietary surveys. Na contents are usually different in individual manufactured foods and the content may differ according to cooks in restaurant or homemade meals, which causes the difference between actual intakes and calculated results.
L100-103 “Dietary records may thus,,,,, but they have a high burden and can misrepresent dietary habits due to behavioral alteration”
One of another negatives is that, people tends to omit complicated foods they actually had from the record to avoid the bothersome.
We thank the reviewer for these remarks. We included the risk of measurement errors based on database inaccuracies as a general limitation for all dietary assessments and underreporting of complicated meals as limitation of diet records.
Line 66
‘All dietary assessment tools rely on food databases, thus database inaccuracies (e.g. differences in nutrients between processed, restaurant cooked and home cooked foods) can result in measurement errors.’
Line 112
‘To lower burden, respondents may simplify or underreport complicated meals that they consumed. Further, keeping a diet record of foods may alter eating behaviors in and on itself, such as opting for healthier and more favorable foods.’
L171 “During terrestrial space flight”
What is the rationale in discussing the special environment in this manuscript discussing preferred method for use in clinical settings.
A major strength of this study is the extremely controlled environment and diet: it provides a valid high-quality reference standard for sodium and potassium intake, which is something that many other studies that investigate sodium intake estimation lack. We do realize that the results from the simulated Mars trip may not be generalizable to the population at large. We did already bring up that the results may not be generalizable to the CKD population, but we will expand this further to the general population and point out that this is a limitation.
Line 216
‘A major strength of this study is the controlled environment and diet: it provides a valid high-quality reference standard for sodium and potassium intake. However, this setting limits the generalizability to free living individuals with a variable intake of sodium and potassium. Furthermore, only young healthy men were included in this study, thus More importantly, it is currently it is unknown whether these infradian rhythms in aldosterone and cortisol affect sodium and potassium balance in women and/or CKD patients to a similar extent.
L217 “Figure 1” should be “Figure 2”
I read the reference [28], but I could not find the figure 1C (single vs 15 years) in the reference.
We want to thank the reviewer for pointing this out. Two figures were named figure 1, we changed one into figure 2 accordingly.
In the study that is discussed, sodium intake was estimated with a single baseline collection and the average of samples collected during a 1-, 5-, and 15-year follow-up. It is correct that this figure is not presented in the study paper. We used original data from this study in the study to create new figures.
Line 225
‘When estimating dietary sodium intake with multiple 24-hour urine collections instead of a single 24-hour urine collection, estimated sodium intake was >0.8 g different in half of the subjects (Figure 21B). The fact that estimated population sodium intake was similar with both methods, points out that the inaccuracy is particularly found when estimating individual sodium intake (Figure 21A).’

Reviewer 2 Report
Some reviews already discussed about the difference in risk of Na consumption on CVD mortality or incidence in which Na consumption was measured by 24hour urines or spot urines shown in epidemiological cohort studies. In my understanding, there have been few manuscript discussing preferred methods focusing urinary measurement, rather than dietary survey methods, to estimate Na and K intakes in clinical settings giving dietary advices.
In this current manuscript, authors seems to try to focus on better assessment method to use in those clinical settings although it is not clear in the title. But it is obvious from the abstract in which authors emphasize the importance of the accuracy of Na intake estimation and without it, clinicians and dieticians may provide misdirected advices. And they compared Na and K estimates obtained from various urine specimens and various dietary surveys.
If that (what is the preferable method to estimate Na and K in clinical settings) is the idea of the authors, my idea is that the manuscript lacks requisite considerations.
WHO’s recommendation of salt intake (less than 5g/day) or recommendations by hypertension societies (less than 6g/day in many case) are well below the average habitual consumption level in most of the civilized society with the exception of minority people who do not have habit of use of salt. It is obvious if one see the foods and menus consumed in the countries. So, dieticians working in clinical settings make dietary advises in the assumption that the current habitual salt intake level would not be lower than the salt reduction target values. I think it is not harmful for the patients, because the target values are not upper limit of requirement but upper limit of harmful level.
In clinical settings, it is more important that dieticians can make advises knowing what are the main source foods of Na or K in the patient’s diet and their contributions to the daily intake. Without it, dieticians cannot make suitable advice. In this point, dietary surveys (FFQ, 24hour dietary recall, or dietary records) have significant advantages over estimates using urine specimen. Authors do not discuss about this point, therefore, I’m afraid clinicians and dieticians who seek for better Na or K estimate method for use in dietary education settings may not find useful knowledge from this manuscript. They would want methods in which they can guess the salt intake trend of a patient, upward or down ward, and the contributing foods.
As a whole, authors discussed about “accuracy” of Na intake estimation derived from various urine specimen. Discussions over effect of salt reduction by diet education programs using various urine or dietary assessment, if they wanted to discuss on the preferred estimation method.
Minor points
L46 “Accurate estimation”
I’m afraid authors use the word “accurate” in a rather crude manner. I think the accuracy desired in the clinical settings would be how the habitual salt intake level of a participant can be estimated.
L60 “Dietary assessment tools”
Food data base is the important point in assessing Na intake from dietary surveys. Na contents are usually different in individual manufactured foods and the content may differ according to cooks in restaurant or homemade meals, which causes the difference between actual intakes and calculated results.
L100-103 “Dietary records may thus,,,,, but they have a high burden and can misrepresent dietary habits due to behavioral alteration”
One of another negatives is that, people tends to omit complicated foods they actually had from the record to avoid the bothersome.
L171 “During terrestrial space flight”
What is the rationale in discussing the special environment in this manuscript discussing preferred method for use in clinical settings.
L217 “Figure 1” should be “Figure 2”
I read the reference [28], but I could not find the figure 1C (single vs 15 years) in the reference.
Author Response

(The authors gave the same response as above.)

Round 2
Reviewer 2 Report
As I commented before, it seems authors focused more on the preferable Na/K estimation method in clinical settings, rather than epidemiological cohort and cross sectional study (including validation study), which is clear from the sentences in the abstract, in which the authors use "guidelines", "patients", "clinicians", and "advice", in at least 9 lines out of 15 lines of the abstract. However, authors mainly discuss on “accuracy”, “misclassification”, or “burden” of salt estimate methods. These might be the matter of epidemiology or physiology. I doubt if these are the only conditions necessary for clinical settings.
All through the manuscript, authors seem to mix the features expected for Na/K estimation method in epidemiological study and clinical settings.
I think focusing on “Na/K estimates from urine and future outcomes” or "nature of urinary excretion of Na/K" would be better. I don’t see why authors discuss on “use in clinical settings” without sufficient references. I think references over what information clinicians or dieticians need when they perform dietary education, or efficacy of Na/K estimate method when used in dietary education program (not in epidemiological studies), would be necessary, in the current form of the abstract.
In the last sentence of the abstract, “we explore alternative methods that may improve test accuracy without increasing (patient, in the last sentence of the introduction section) burden”. Please describe the favorable “alternative methods” in the conclusion section.
